# Complete Chloroplast Genome Determination of *Ranunculus sceleratus* from Republic of Korea (Ranunculaceae) and Comparative Chloroplast Genomes of the Members of the *Ranunculus* Genus

**DOI:** 10.3390/genes14061149

**Published:** 2023-05-25

**Authors:** Kang-Rae Kim, So Young Park, Heesoo Kim, Jeong Min Hong, Sun-Yu Kim, Jeong-Nam Yu

**Affiliations:** Animal & Plant Research Department, Nakdonggang National Institute of Biological Resources, Sangju 37242, Republic of Korea; kimkangrae9586@gmail.com (K.-R.K.); cindysory@nnibr.re.kr (S.Y.P.); heesookim@nnibr.re.kr (H.K.); hjmjulia@nnibr.re.kr (J.M.H.); ksuny007@nnibr.re.kr (S.-Y.K.)

**Keywords:** next-generation sequencing, positive selection, DNA barcode, specific barcode, chloroplast genome, *Ranunculus sceleratus*

## Abstract

*Ranunculus sceleratus* (family: Ranunculaceae) is a medicinally and economically important plant; however, gaps in taxonomic and species identification limit its practical applicability. This study aimed to sequence the chloroplast genome of *R. sceleratus* from Republic of Korea. Chloroplast sequences were compared and analyzed among *Ranunculus* species. The chloroplast genome was assembled from Illumina HiSeq 2500 sequencing raw data. The genome was 156,329 bp and had a typical quadripartite structure comprising a small single-copy region, a large single-copy region, and two inverted repeats. Fifty-three simple sequence repeats were identified in the four quadrant structural regions. The region between the *ndhC* and *trnV-UAC* genes could be useful as a genetic marker to distinguish between *R. sceleratus* populations from Republic of Korea and China. The *Ranunculus* species formed a single lineage. To differentiate between *Ranunculus* species, we identified 16 hotspot regions and confirmed their potential using specific barcodes based on phylogenetic tree and BLAST-based analyses. The *ndhE*, *ndhF*, *rpl23*, *atpF*, *rps4*, and *rpoA* genes had a high posterior probability of codon sites in positive selection, while the amino acid site varied between *Ranunculus* species and other genera. Comparison of the *Ranunculus* genomes provides useful information regarding species identification and evolution that could guide future phylogenetic analyses.

## 1. Introduction

The family Ranunculaceae is distributed worldwide, comprising 59 genera and more than 2500 species [1]. Among them, *Ranunculus* (buttercups) is the largest genus, comprising approximately 600 species, and is classified into 20 taxonomic sections [2,3]. *Ranunculus* inhabits tropical and temperate regions within temperate forests, arctic tundra, mountainous regions, freshwater systems, and terrestrial ecosystems [3,4]. Genetically, this genus is diverse with a wide distribution, making its classification highly complex [1]. In particular, taxonomic difficulties have occurred due to the morphological and ecological diversity of traits that have arisen during the evolution of *Ranunculus* in different environments [5,6].

*R. sceleratus* L. is an annual or perennial herbaceous plant in the family Ranunculaceae [7] that primarily inhabits wetlands or slow rivers and is distributed worldwide, including throughout Republic of Korea [8]. Dried and fresh *R. sceleratus* is reportedly used in the treatment of esophageal and breast cancer, as well as malaria-induced jaundice [8,9]. Additionally, as a medicinal plant, it can suppress the occurrence of degenerative diseases by acting as an antibacterial agent and antioxidant [10]. Indeed, plants of the genus *Ranunculus* have been reported to protect the liver and exhibit anti-inflammatory and anti-malarial effects [11,12,13,14,15,16]. They have pharmacological activity and potentially high value as new drug candidates [17]. However, the application of *R. sceleratus* as a drug candidate species has been limited by the reported variation in the major pharmacologically active ingredients within the plant based on geographical differences [18,19].

Besides its medicinal value, *R. sceleratus* has potential value in plant purification and landscaping. *R. sceleratus* has attractive characteristics, including pale yellow flowers and cylindrical fruits during flowering [7]. These features are found around parks, cities, and rivers and provide potential value for attractive landscaping [7]. In particular, this plant exhibits significant potential to purify contaminated sewage by adsorbing heavy metals, such as Fe and Zn, and removing organic matter, such as total nitrogen and phosphorus [20,21,22]. As such, it is important to be able to accurately identify the distinct geographical features as well as the different species within *Ranunculus* to improve their application in the fields of pharmaceuticals and heavy metal decontamination.

To address these gaps in taxonomic and species identification, we previously sought to identify *Ranunculus* species using molecular biological methods [6]. Results showed that species of *Batrachium*, a subgenus of *Ranunculus*, can be identified based on chloroplast or nuclear markers, such as *trnH-psbA*, *matK*, and ITS; however, close relatives, including *Ranunculus peltatus* and *Ranunculus penicillatus*, could not be similarly distinguished [6]. Thus, a marker with high species resolution is required to effectively identify related species of the genus *Ranunculus*, which are difficult to morphologically identify.

As a molecular marker in plants, the mitochondrial genome does not function as a species identification DNA barcode owing to the low level of nucleotide substitutions [23,24]. Therefore, chloroplast (cp) genomes are commonly assessed to achieve the evolutionary and species identification of plants [25,26,27]. In particular, the *trnH-psbA*, *matK*, *rpoB*, *rpoC1*, and *trnL-F* intergenic spacer regions of the chloroplast genome region are commonly employed; however, a consensus has not been reached regarding universally available barcodes because of discrepancies in each plant taxon [25,28]. Hence, while DNA barcodes of these chloroplast sequence regions are useful for phylogenetic and barcode studies at high taxonomic levels, they are not suitable at lower taxonomic levels owing to insufficient variation [29]. Accordingly, specific barcodes for lower taxonomic levels are required [29].

Specific barcodes examine repeats, indels, and substitutions in the chloroplast genome for species identification and hotspot regions to identify loci that represent species-level differences. As such, these barcodes represent a powerful tool for the identification of species that cannot be identified by commonly used barcodes [27]. Moreover, optimized primers for specific barcodes are needed to improve polymerase chain reaction (PCR) efficiency and avoid the risk of ambiguous alignment by loci [30].

Currently, studies on the genus *Ranunculus* have reported the complete cp genomes for *Nuphar advena* and *Ranunculus macranthus* in angiosperms [26,31,32,33,34]. However, comparative cp genome studies for *Ranunculus* have not been conducted. These studies have important implications for understanding interspecies phylogeny and for species identification and evolutionary studies.

This study aimed to (1) sequence the complete cp genome of *R. sceleratus* from Republic of Korea and compare it with *R. sceleratus* from China to identify population-specific regions, (2) screen potential barcode markers for specific barcodes by exploring phylogenetic positions for *R. sceleratus* from Republic of Korea and hotspot regions via comparison of 12 cp *Ranunculus* genomes, and (3) explore positive selection in *Ranunculus* to advance the current genetic and evolutionary understanding of this genus.

## 2. Materials and Methods

### 2.1. Plant Sampling and Sequencing

The leaves of *R. sceleratus* were collected from Yeongi-gun, Chungcheongnam-do, Republic of Korea (36°37′32″ N, 127°17′16″ E). The identification and collection of plants were conducted by the author Sun-Yu Kim. The voucher number was NNIBRVP70282, which was deposited with the Nakdonggang National Institute of Biological Resources. Total genomic DNA extraction was performed on *R. sceleratus* leaves with the DNeasy^®^ Plant Mini kit (Qiagen) according to the manufacturer’s instructions. An Illumina HiSeq 2500 sequencing library was prepared according to the manufacturer’s protocol. The DNA library was sequenced with a 150 bp paired-end.

### 2.2. Chloroplast Genome Assembly and Annotation

A short raw read was assembled to obtain the *R. sceleratus* cp genome, which was then assembled using NOVOPlasty ver. 4.3.1 software [35]. The cp genome was verified for sequence identity using GetOrganelle ver. 1.7.7 [36]. Both assemblers confirmed the sequence of the same cp genome. The complete *R. sceleratus* cp genome was annotated using CPGAVAS2 [37]. *Ranunculus seleratus* from Republic of Korea was annotated using *Ranunculus seleratus* from China (MK253452). Areas with errors in the annotations were manually corrected using Geneious ver. 11.0.1 software. Finally, the annotated cp genome sequence was uploaded to GenBank under the accession number ON755204 using NCBI BankIt. An image of the cp genome for *R. sceleratus* from Republic of Korea was drawn using OrganellarGenome DRAW [38].

### 2.3. Simple Sequence Repeat Analysis

The simple sequence repeat (SSR) regions in the cp genome were screened using the MIcroSAtellite (MISA) tool [39]. The SSR motif conditions were 10, 5, 4, 3, 3, and 3 for mono-, di-, tri-, tetra-, penta-, and hexa-nucleotides, respectively. Additionally, duplicates and errors were manually screened for SSR regions.

Repeat Finder of Geneious ver. 11.0.1 was used to identify the types of long repeat sequences, i.e., palindromic, forward, reverse, and complement. Repeat sequence identification conditions were set to a minimum repeat size of 20 bp and 100% sequence identity.

### 2.4. Phylogenetic Analysis

To construct phylogenetic trees, 13 cp genomes of the species belonging to the family Ranunculaceae were downloaded from NCBI GenBank. The dataset was constructed by arranging the genome and gene sequences in the same order. The datasets were then realigned using MAFFT ver. 7.490 [40], and a phylogenetic tree was constructed using two methods: maximum likelihood (ML) and Bayesian inference (BI). The ML method used jModelTest ver. 2.1.7 [41] to select the appropriate optimal model (GTR + G + I) from the dataset and constructed the phylogenetic tree using PhyML ver. 3.0 [42]. Moreover, 1000 bootstrap replicates were included for the ML tree. The BI method reconstructed a phylogenetic tree using MrBayes ver. 3.2.7 [43] under the GTR + G + I model (lset nst = 6, Rates = invgamma), which is the optimal base substitution model. The Markov Chain Monte Carlo (MCMC) algorithm used in the analysis was repeated for 10 million MCMC and sampled every 100 generations, with the first 25% discarded as burn-in. The posterior probability values for each node were determined by consensus on multiple trees. Additionally, the phylogenetic trees drawn by both methods were constructed by consensus. The support for the ML and BI consensus trees was expressed as bootstrap values and posterior probabilities, respectively.

### 2.5. Comparative CP Genome Analysis and Screening of Barcoding Regions

Chloroplast genome sequences for *Ranunculus* species were downloaded from NCBI (Appendix A) and rearranged using MAFFT ver. 7.490 [40]. For the aligned sequence, the coding sequence (CDS) present for each species was extracted using Geneious ver. 11.0.1 [44]. Additionally, the intergenic regions in 12 cp genomes were extracted and sequenced. To analyze the nucleotide diversity (Pi), the CDS and intergenic regions of the cp genome for *Ranunculus* were analyzed using DnaSP ver. 6.12 software [45]. *Ranunculus* species genomes were compared using mVISTA software (http://genome.lbl.gov/vista/index.shtml, accessed on 2 March 2022) and visualized in shuffle-LAGAN mode.

### 2.6. Specific Barcode Region Primer Design

Primers for effective molecular markers were designed based on the identifiable intergenic region of *Ranunculus* species. First, *Ranunculus austro-oreganus* L.D. Benson, *R. bungei* Steud., *R. cantoniensis* DC., *R. japonicas* Thunb., *R. macranthus* Scheele, *R. occidentalis* Nutt., *R. pekinensis* (L.Liou) Luferov, *R. reptans* L., *R. repens* L., *R. sceleratus* L., and *R. yunnanensis* Franch were downloaded from GenBank. Additionally, a dataset including the cp genome of *R. sceleratus* from Republic of Korea assembled in this study was used for subsequent analysis. A total of 12 cp genomes were multi-aligned using MAFFT ver. 7.490 [40]. Among the DNA barcode regions (coding sequence and intergenic), 13 CDS (Pi ≥ 0.02) and 48 intergenic regions (Pi ≥ 0.040) with high nucleotide diversity were selected as hotspots. The PCR amplified product size range was 150–1500 bp.

To evaluate the success of species identification for specific barcode markers, two methods were employed, namely, the phylogenetic tree-based method and the sequence similarity-based method. The phylogenetic tree method analyzed two phylogenetic trees (32 specific barcode region sequences), ML and BI, and judged whether they were consistent with the tree drawn based on the cp genome. The similarity-based method constructed a local database using BLAST 2.2.29+; and the sequences were run through BLAST to perform the query. Identification of species barcoding markers was considered successful when the concordance for homogeneity was 100%.

### 2.7. Positive Selection Analysis

Extraction of the aligned CDS for *Ranunculus* was performed using Geneious ver. 11.0.1 [44]. For positive selection analysis, the optimized branch site model method was applied using the CODEML program of the PAML ver. 4.10.6 package [46,47]. We implemented Bayes Empirical Bayes (BEB) and Naive Empirical Bayes (NEB) methods to identify specific amino acid sites in potential positive selection genes to calculate posterior probabilities. Codon sites with high posterior probability (*P* > 0.5) were considered positively selected sites [48,49]; positive selection: ratio ω > 1, neural selection: ω = 1, and negative selection: ω < 1 [50]. An alternative branching site model (model = 2, NSsites = 2, Fix = 0) and a neutral branching site model (model = 2, NSsites = 2, Fix = 1, Fix ω = 1) were applied to identify positively selected sites. Genes with a *p*-value < 0.05 and a positively selected site were considered a positive selection. Visualization of the amino acid sequence portions of positively selected genes was performed using Jalview ver. 2.11.2 software [51].

## 3. Results and Discussion

### 3.1. CP Genome Characterization for Ranunculus Species

Whole-genome sequencing of *R. sceleratus* from Republic of Korea produced 958 million reads using the HiSeq 2500 platform, yielding a total of 144 GB (Table 1). The complete cp genome of *R. sceleratus* from Republic of Korea was obtained by assembly based on the reference genome of *R. sceleratus* from China (NCBI Accession No.: MK253452). The cp genome of *R. sceleratus* from Republic of Korea was 156,329 bp in length and had a typical quadripartite structure (Figure 1). The cp genome contained large single-copy (LSC, 85,840 bp), small single-copy (SSC, 19,885 bp), and inverted repeat (IR) regions; the IR comprised two copies of IRs a and b and was 25,302 bp in size (Table 1).

Although the cp genome of higher plants is highly conserved, the extension and contraction between the IR, LSC, and SSC regions cause differences in genome length between species [26,33,34,52,53]. In the case of *Ranunculus* species, a change in gene position due to the expansion and contraction of IR and SSC was observed. More specifically, in nine species (except *R. austro-oreganus* and *R. occidentalis*), the *atpH* gene was located between the *atpF* and *atpI* genes in the 1.3 kb LSC region (Figure 2). In contrast, the *atpH* gene in *R. austro-oreganus* and *R. occidentalis* was located between the *ycf1* and *trnN-GUU* (11.5 kb) genes in the SSC region. Structural variations in IR and SSC can result in gene rearrangement [54,55]. In this study, the SSC region lengths of *R. austro-oreganus* (21,249 bp) and *R. occidentalis* (21,269 bp) were extended by more than ~2 kb compared to the SSC region lengths of *R. japonicus* and *R. macranthus* (18,909 bp) and, thus, might represent a major cause of *atpH* gene rearrangement [54,55].

No contraction or expansion was detected in the IR and SSC regions between *R. sceleratus* from Republic of Korea and China; however, a small 5 bp sequence difference was observed in the LSC region (Appendix A). Given that the cp genome tends to be highly conserved across species [56,57,58], it was estimated that low levels of sequence variation exist between the Republic of Korean and Chinese species. Hence, these minor sequence differences in the cp genome serve as useful markers for differentiating geographic populations [27]. This marker can potentially be used for population identification, but further population studies are needed to verify this.

The GC content of the cp genome was 37.9%, which corresponded with that of *R. sceleratus* from China; 12 other *Ranunculus* species genomes were reported to have a similar GC content (37.7–37.9%; Appendix A). The GC content of the LSC region of the 13 cp genome was 35.9–36.2%; that of the SSC region was 31.0–31.7%, and that of the IR region was 42.8–43.6%, with the largest difference in GC content in the IR region. The cp genome was classified into self-replication, photosynthesis, and other genes, as it is in other plants (Appendix A). The *Ranunculus* species, excluding *R. austro-oreganus*, *R. occidentalis*, and *R. sceleratus* from China, carried 112 genes. The genetic and structural makeup of most species was similar. That is, *R. sceleratus* from Republic of Korea and China carried *infA*, while *R. austro-oreganus* and *R. occidentalis* had *ycf15* genes (Appendix A). Notably, the *infA* gene has been lost in many angiosperms, and in certain plants, it has been transferred to the nucleus [59,60]. Loss of the genome may be a low-cost strategy for rapid replication in adverse environmental conditions [61].

### 3.2. SSR Analysis of Ranunculus CP Genomes

#### 3.2.1. Long Repeats

Long repeats are important hotspots for genetic rearrangement and population variation in the cp genome [62,63,64]. A total of 402 repeats were identified for 12 *Ranunculus* cp genomes (Appendix A), including 148 forward repeats, 127 reverse repeats, 96 complementary repeats, and 31 palindromic repeats. Regarding the number of repeats by species, *R. japonicus* contained the most (64), while *R. yunnanensis* had the fewest (18; Figure 3). According to the number of iterations, the largest number of species contained 42 repeats (Figure 4, Appendix A). Notably, most species could be identified based on the total number of repeats, excluding *R. austro-oreganus* (31), *R. occidentalis* (31), *R. bungei* (34), and *R. reptans* (34; Appendix A). However, species with the same number of repeats could be identified based on the type of repeats and the ratio of repeats between species (Appendix A). Repeat sequences rearrange the cp genome [63] and reportedly enable interspecies identification owing to their random distribution [63,65].

#### 3.2.2. SSR Analysis

In this study, 53, 52, 47, 43, 45, 47, 48, 45, 49, 38, 44, and 37 SSRs were identified for *R. sceleratus* from Republic of Korea, *R. sceleratus* from China, *R. austro-oreganus*, *R. bungei*, *R. cantoniensis*, *R. japonicus*, *R. macranthus*, *R. occidentalis*, *R. pekinensis*, *R. reptans*, *R. repens*, and *R. yunnanensis*, respectively (Appendix A). The motif distribution of the *Ranunculus* species was 58.21% mono-nucleotide, 17.34% di-nucleotide, 15.69% tetra-nucleotide, 6.02% tri-nucleotide, 2.37% penta-nucleotide, and 0.36% hexa-nucleotide (Appendix A). The number of SSRs throughout the genome is generally large in the order of di-, mono-, tri-, and tetra-nucleotides [66,67,68]. However, herein, the SSR distribution in the cp genome was large in the order of mono-, di-, tetra-, and tri-nucleotides, with a difference observed in the distribution of SSR between the genome and cp genome. This appears to be a feature of SSR distribution in the cp genome of plants [69,70].

In the cp genome of *R. sceleratus* from Republic of Korea, 64.15% (34 SSRs) of the SSRs were located in the LSC region, followed by 28.30% (15 SSRs) in the SSC region and 7.55% (4 SSRs) in the IR (Appendix A). The *R. sceleratus* from Republic of Korea and China exhibited small intraspecies sequence differences in the LSC region, which was located between the *ndhC* and *trnV-UAC* genes. However, an indel difference in the TAAAG repeat sequence was detected (Appendix A). Hence, the region between these two genes may be useful as a genetic marker to distinguish *R. sceleratus* species from Republic of Korea and China.

The number of mono-, di-, and tetra-nucleotide repeats was highly variable in the cp genome of *Ranunculus* species. In fact, if the total number of SSRs among species was the same, it was possible to distinguish between species based on differences in SSR distribution (*R. austro-oreganus*: 47, *R. japonicus*: 47, *R. cantoniensis*: 45, and *R. occidentalis*: 45). SSR variations in the cp genomes of nine *Ranunculus* taxa were determined to be useful for comparing phylogenetic relationships through genetic polymorphisms at the population and species level, suggesting their importance for SSR studies.

### 3.3. Phylogenetic Analysis

Datasets used for phylogenetic analysis were classified and constructed to include the cp genome, coding sequences, intergenic regions, and specific barcode regions. ML and BI tree analysis of the four datasets showed that *Ranunculus* was generated as a single clade; the bootstrap value was strongly supported at 100% for all clades (Figure 5). The first clade comprised six species (*R. austro-oreganus*, *R. occidentalis*, *R. japonicus*, *R. cantoniensis*, *R. macranthus*, and *R. repens*), while the second comprised *R. sceleratus* from China (MK253452), *R. sceleratus* from Republic of Korea (ON755204), *R. bungei*, *R. pekinensis*, and *R. yunnanensis*. *Ranunculus sceleratus* from Republic of Korea was most closely related to *R. sceleratus* from China.

The complete cp genome has been proposed to be useful for taxonomic reconstruction [26,32] and utilized in previous studies to resolve interspecies phylogenetic relationships within *Ranunculus*. The current study also demonstrated the potential of using the complete cp genome to resolve phylogenetic relationships [4,26,32,71]. Although the cp genome is phylogenetically informative, the closely related *R. austro-oreganus*, *R. occidentalis*, *R. bungee*, and *R. pekinensis* required high-resolution specific barcodes for species identification.

The CDS phylogenetic tree did not clearly differentiate between these four closely related species; however, the intergenic regions and 16 specific barcode regions facilitated their identification. In particular, the tree constructed based on specific barcode regions efficiently differentiated the species using a small number of intergenic regions.

### 3.4. Barcoding Region Screening of Genome Divergence Regions

Genome comparison plots were created using mVISTA to assess the sequence similarity of the cp genomes between *Ranunculus* species (Appendix A). The aligned gene positions among *Ranunculus* species showed that the positions and sequences of genes, except *atpH*, were conserved. When comparing the same species, relatively no difference was detected in the sequences (*R. sceleratus* from Republic of Korea and China); however, a clear difference in sequences was observed between *Ranunculus* species. Notably, the IR was relatively more conserved than the LSC and SSC regions, which agrees with results for other plant species.

We multi-aligned the cp genome sequence of the *Ranunculus* genus and screened the protein-coding regions, intergenic regions, and intronic regions to identify unique regions within the genome. DnaSP analysis searched for polymorphic genes in coding sequences and intergenic regions. The nucleotide diversity (Pi) values ranged from 0.000 (*rpl23*) to 0.043 (*ccsA*). The most polymorphic genes were *ccsA*, *matK*, *rpl32*, *rps3*, *ndhE*, *rps15*, *ndhG*, *ndhA*, *psbH*, *accD*, *clpP*, *atpF*, and *ndhH* (Pi ≥ 0.02), and the least polymorphic gene was *rpl23* (Pi: 0.000; Figure 6). The nucleotide diversity of the intergenic region ranged from 0.002 (*rpl23-trnL*-*CAU*) to 0.103 (*ndhF-rpl32*).

The highly polymorphic intergenic regions were *ndhF-rpl32*, *ccsA-ndhD*, *petG-trnW-CCA*, *trnH-GUG-psbA*, *rpl32-trnL-UAG*, *psbT-psbN*, *rpl16-rps3*, *trnL-UAG-ccsA*, *trnW-CCA-trnP-UGG*, *rps8-rpl14*, *ndhE-ndhG*, *rps16-trnQ-UUG*, *psaC-ndhE*, *ndhD-psaC*, *petA-psbJ*, *psaJ-rpl33*, *ndhG-ndhI*, *rps15-ycf1*, *petN-psbM*, *petD-rpoA*, *trnS-GGA-rps4*, *trnD-GUC-trnY-GUA*, *accD-psaI*, *ndhH-rps15*, *trnG-GCC-trnfM-CAU*, *atpF-atpI*, *trnG-UCC-trnR-UCU*, *psbI-trnS-GCU*, *trnT-UGU-trnL-UAA*, *psbZ-trnG-GCC*, *petL-petG*, *trnK-UUU-rps16*, *ndhC-trnV-UAC*, *rpl36-rps8*, *psbK-psbI*, *rps19-rpl2*, *rpl14-rpl16*, *rps2-rpoC2*, *ycf3-trnS-GGA*, *psbA-trnK-UUU*, *rpl22-rps19*, *trnC-GCA-petN*, *trnT-GGU-psbD*, *rps18-rpl20*, *psbE-petL*, *psbC-trnS-UGA*, *rpl33-rps18*, and *trnP-UGG-psaJ* (Pi ≥ 0.040).

Herein, when the sequence diversity of 11 *Ranunculus* species was compared, the noncoding region was more variable than the coding region (Figure 6). Similarly, in angiosperms, the nucleotide diversity is higher within the intergenic region than within the coding region [27,72,73,74]. Certain hotspots within these variable regions were used to design primers to differentiate *Ranunculus* species from others, demonstrating that these regions can be utilized as potential barcoding markers. More specifically, within the protein-coding region, 13 regions with a Pi ≥ 0.02 and 48 intergenic regions with a Pi ≥ 0.040 were set as hotspot regions.

### 3.5. Barcode Validation of Hotspot Regions in the Genome

Among the 48 hotspot regions, those useful as specific barcodes were selected based on BLAST and phylogenetic tree analyses. The specific barcode regions *ndhG-ndhI*, *petN-psbM*, *atpF-atpI*, *ndhC-trnV-UAC*, *trnT-GGU-psbD*, and *psbE-petL* had a BLAST-based species identification rate of 100%. For the remaining 10 markers, the species identification rate using BLAST was 83.3–91.7% (Table 2).

Based on ML and BI analysis, the single hotspot region markers *ndhC-trnV-UAC* and *psbE-petL* had 100% species identification, followed by *rps8-rpl14*, *petN-psbM*, *atpF-atpI*, and *trnT-GGU-psbD* with 91.7% and *rpl32-trnL-UAG*, *rpl16-rps3*, *rps16-trnQ-UUG*, *ndhG-ndhI*, *accD-psaI*, *trnG-GCC-trnfM-CAU*, *trnT-UGU-trnL-UAA*, *psbZ-trnG-GCC*, and *trnK-UUU-rps16* with 83.3% (Appendix A). The *petG-trnW-CCA* had the lowest species identification rate at 75% and did not match the phylogenetic tree for other intergenic markers. Indeed, if the evolution and gene locus of a species differ, then the gene tree and the species tree may differ [75].

With the exception of *R. austro-oreganus* and *R. occidentalis*, the first group of markers to distinguish between nine *Ranunculus* species included *trnT-UGU-trnL-UAA* and *trnK-UUU-rps16*. The second group of markers to distinguish between nine *Ranunculus* species, with the exception of *R. bungei* and *R. pekinensis*, included *petG-trnW-CCA*, *rpl32-trnL-UAG*, *rpl16-rps3*, *rps16-trnQ-UUG*, *accD-psaI*, *trnG-GCC-trnfM-CAU*, and *psbZ-trnG-GCC*. With the exception of *R. macranthus* and *R. repens*, the third group of markers to distinguish between nine *Ranunculus* species included *rps8-rpl14*. Combining two or more markers provides a higher rate of species identification than a single marker [76,77]. Therefore, to enhance species identification, ML and BI phylogenetic tree construction and BLAST were performed to analyze the species identification rate by combining the first, second, and third group markers.

Each of the 37 datasets showed a high species identification rate of 100% compared to the existing single markers (Appendix A). The hotspot markers were *petG-trnW-CCA + trnT-UGU-trnL-UAA + rps8-rpl14*, *rpl16-rps3 + rps8-rpl14*, *rpl16-rps3 + trnT-UGU-trnL-UAA + rps8-rpl14*, *rpl16-rps3 + trnK-UUU-rps16 + rps8-rpl14*, *rps16-trnQ-UUG + rps8-rpl14*, *accD-psaI + rps8-rpl14*, *accD-psaI + trnT-UGU-trnL-UAA + rps8-rpl14*, *accD-psaI + trnK-UUU-rps16 + rps8-rpl14*, *trnG-GCC-trnfM-CAU + rps8-rpl14*, *trnG-GCC-trnfM-CAU + trnT-UGU-trnL-UAA + rps8-rpl14*, *trnG-GCC-trnfM-CAU + trnK-UUU-rps16 + rps8-rpl14*, *trnT-UGU-trnL-UAA + rps8-rpl14*, *psbZ-trnG-GCC + rps8-rpl14*, *psbZ-trnG-GCC + trnT-UGU-trnL-UAA + rps8-rpl14*, *psbZ-trnG-GCC + trnK-UUU-rps16 + rps8-rpl14*, and *trnK-UUU-rps16 + rps8-rpl14*, which showed 100% species identification (Appendix A).

The combination of specific barcode markers identified seven-fold more variant sites than conventional single specific barcode markers. Given that markers have different rates of nucleotide variation at different loci, when assessing closely related species, the application of combination markers may be more advantageous for species identification [78]. The conventional barcoding markers, *rbcL*, *psbA-trnH-GUG*, and *trnL-UAA-trnF-GAA*, showed a low 75% species identification rate using BLAST and phylogenetic tree-based methods (Table 3). Therefore, the combination of specific barcode markers and single specific barcode markers with high species resolution developed in this study can replace the existing barcoding markers.

Primers were designed for the flanking regions of these markers to enable the use of 16 hotspot regions as specific barcodes (Table 4). The 100% species identifiable markers for *Ranunculus* showed high resolution for genetically closely related species, such as *R. austro-oreganus*, *R. occidentalis*, *R. bungei*, and *R. pekinensis*. Accordingly, these markers can be effectively applied for the identification of very closely related species of the genus *Ranunculus*.

### 3.6. Positive Selection Analyses

Positive selection analysis was performed on 66 CDS. In the likelihood ratio analysis, most *p*-values for all genes were >0.05, i.e., insignificant. In contrast, in the BEB test, *atpF*, *ndhE*, *ndhF*, *rpl23*, *rpoA*, *rps4*, and *ycf4* showed posterior probabilities ≥ 0.7, with *ndhE*, *ndhF*, and *rpl23* having values ≥ 0.9. In the NEB method, positively selected sites in *atpF*, *ndhF*, *rps4*, and *rpoA* genes were detected as ≥0.8 (Appendix A). In a previous study, codon regions with high posterior probability based on BEB and NEB analyses were considered positive selection genes [48]. Therefore, we ultimately selected *atpF*, *ndhE*, *ndhF*, *rpl23*, *rpoA*, and *rps4* as positive selection genes. The six amino acid characteristics between *Ranunculus* species and other genera are shown in Figure 7.

The *atpF* gene, which is required for electron transport and photophosphorylation during photosynthesis, encodes the H^+^-ATP synthetase subunit [79]. The *rpoA* gene encodes a plastid-encoded RNA polymerase (PEP) that regulates gene expression [80,81,82]. The *rpl* and *rps* genes encode large and small subunits of ribosomal proteins and are differentially expressed according to abiotic and biotic environmental factors by regulating plant growth and transcription [83,84].

The chloroplast *ndh* gene forms a thylakoid NADH dehydrogenase (Ndh) complex homologous to the encoded polypeptide and mitochondrial complex I [85,86,87]. As such, *ndh* plays an essential role in photosynthesis in higher plants [88]. In particular, the Ndh complex generated from *ndh* protects against photooxidation-related stress and maintains an optimal photophosphorylation rate [88]. However, the *ndh* gene is presumably not required in mild, unstressed environments and exhibits rapid changes in function under light-related stress [88,89,90]. In this study, *ndhE*, *ndhF*, *rpl23*, *atpF*, *rps4*, and *rpoA* were confirmed to have high positive selection sites. Considering that various light-related stressful environments alter the biochemical composition and morphology of plants during acclimatization [91], it is inferred that these positive selection sites represent mutations in gene function related to environmental resistance as a light-related stress response in *Ranunculus* species. Therefore, genes that are positively selected and adapted to these genes could represent adaptation to new environmental conditions for *Ranunculus* species.

## 4. Conclusions

*Ranunculus sceleratus* (family: Ranunculaceae) is a medicinally and economically important plant; however, gaps in taxonomic and species identification limit its practical applicability. The aim of this study was to elucidate the chloroplast genome of *R. sceleratus* from Republic of Korea, identify how it differed from the chloroplast genome of *R. sceleratus* from China, and identify potential specific barcodes through genome comparisons in the genus *Ranunculus*. The chloroplast genome was assembled from Illumina HiSeq 2500 sequencing raw data. The genome was 156,329 bp and had a typical quadripartite structure comprising a small single-copy region, a large single-copy region, and two inverted repeats. Fifty-three simple sequence repeats were identified in the four quadrant structural regions. The region between the *ndhC* and *trnV-UAC* genes could be useful as a genetic marker to distinguish between *R. sceleratus* populations from Republic of Korea and China. The *Ranunculus* species formed a single lineage. To differentiate between *Ranunculus* species, we identified 16 hotspot regions and confirmed their potential using specific barcodes based on phylogenetic tree and BLAST-based analyses. The *ndhE*, *ndhF*, *rpl23*, *atpF*, *rps4*, and *rpoA* genes were considered positively selected owing to their high posterior probability. The amino acid site varied between *Ranunculus* species and other genera. Comparison of the *Ranunculus* genomes provides useful information regarding species identification and evolution that could guide future phylogenetic analyses.

## Figures and Tables

**Figure 1 genes-14-01149-f001:**
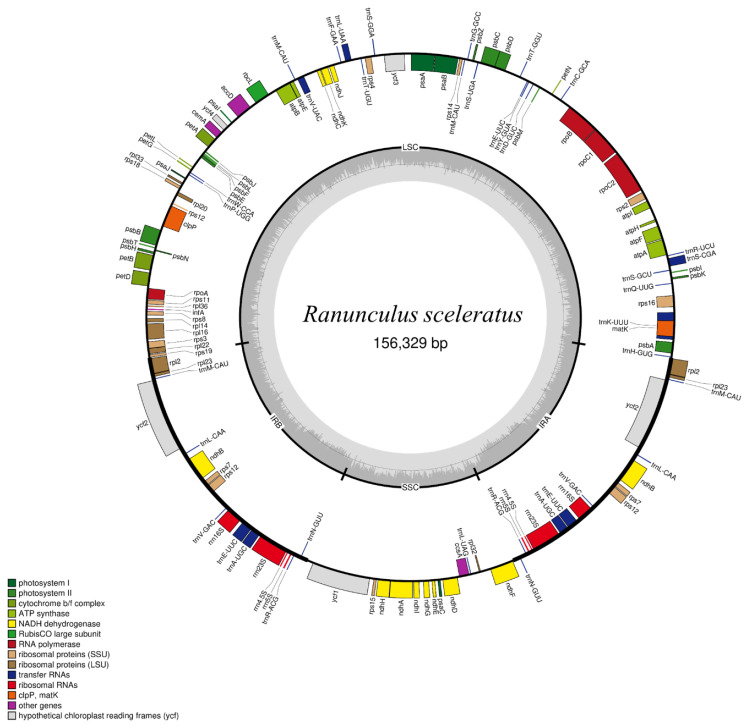
Structure of the complete chloroplast genome of *Ranunculus sceleratus* from Republic of Korea.

**Figure 2 genes-14-01149-f002:**
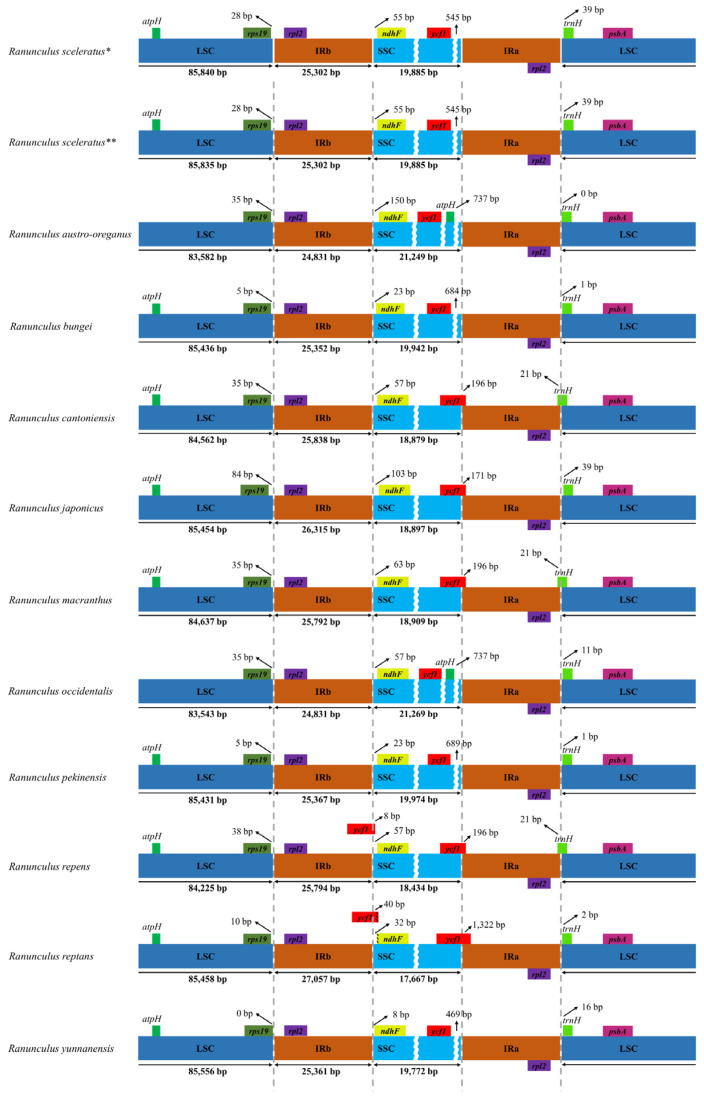
Characteristics of the region adjacent to the contraction and expansion of inverted repeats (*: *R. sceleratus* from Republic of Korea, **: *R. sceleratus* from China).

**Figure 3 genes-14-01149-f003:**
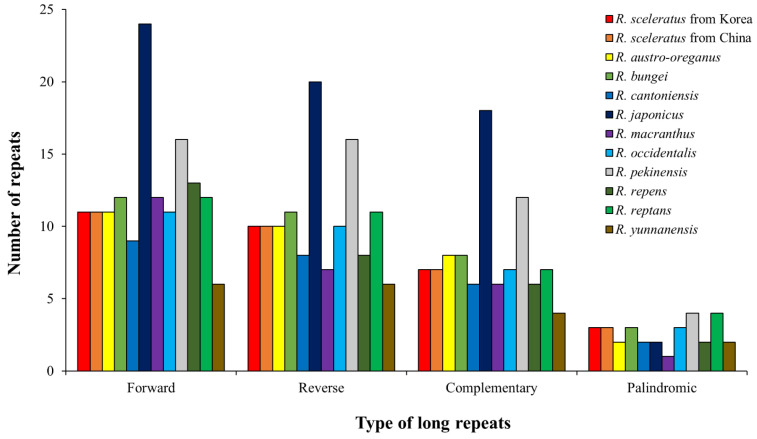
Long repeat type distribution of *Ranunculus* species.

**Figure 4 genes-14-01149-f004:**
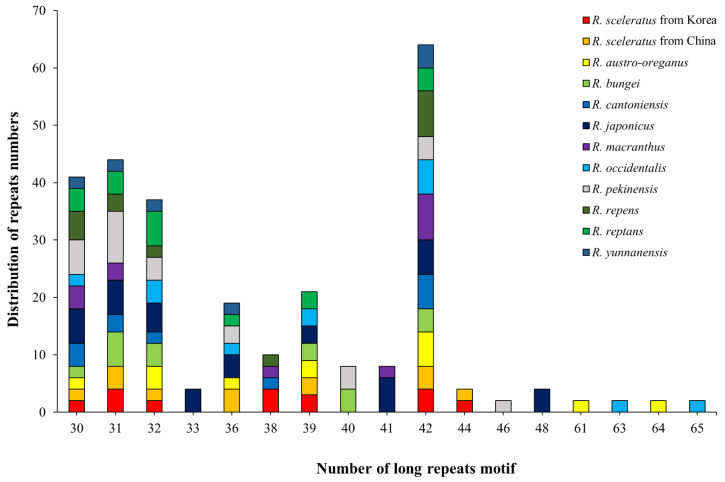
Distribution of species based on the number of long repeats.

**Figure 5 genes-14-01149-f005:**
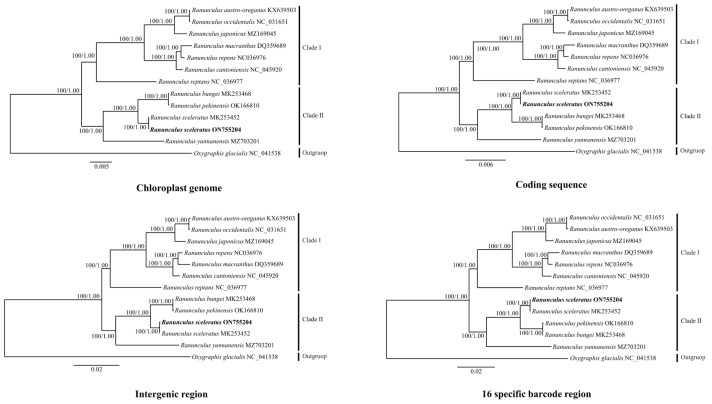
Phylogenetic trees comprising 12 species in the family Ranunculaceae using the maximum likelihood method, based on the chloroplast gene sequence. The numbers above the nodes represent the bootstrap support value for each branch.

**Figure 6 genes-14-01149-f006:**
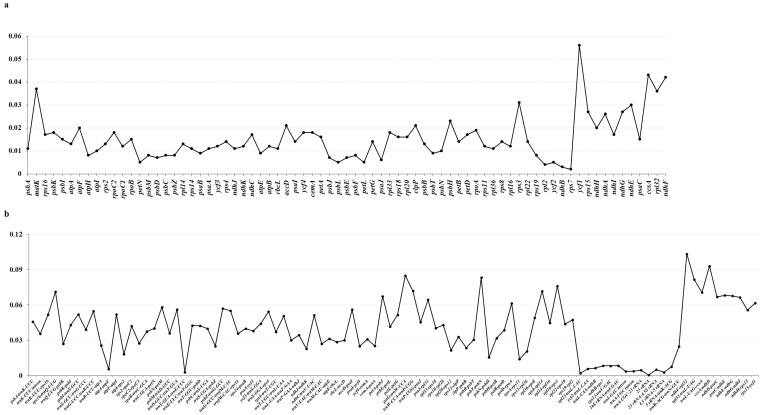
Nucleotide diversity in the entire chloroplast genome of 11 species. (**a**) coding sequence; (**b**) intergenic region.

**Figure 7 genes-14-01149-f007:**
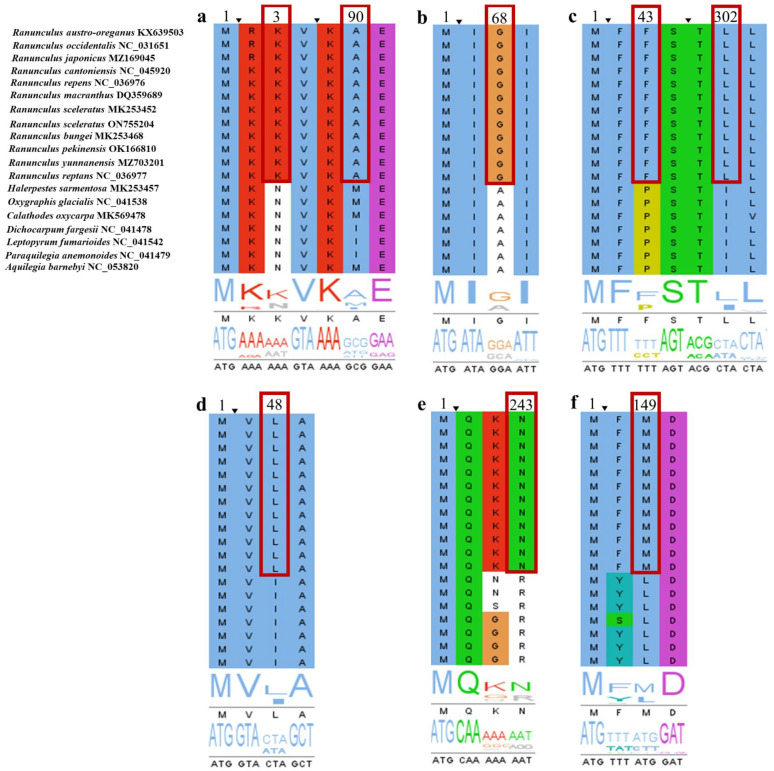
Amino acid sites of positive selection in the genus *Ranunculus* in the family Ranunculaceae. (**a**), *atpF*; (**b**), *ndhE*; (**c**), *ndhF*; (**d**), *rpl23*; (**e**), *rpoA*; (**f**), *rps4*. The red square box represents the variation of amino acid positions in the species of the genus *Ranunculus*.

**Table 1 genes-14-01149-t001:** Chloroplast genome information of *Ranunculus sceleratus* from Republic of Korea.

Total Reads	958,608,148
Percentage of subsampling reads (%)	15.0
GC content of total reads (%)	41.22
Chloroplast genome length (bp)	156,329
Large single-copy length (bp)	85,840
Small single-copy length (bp)	19,885
Inverted repeat length (bp)	25,302
Total number of genes	112

**Table 2 genes-14-01149-t002:** Sequence matching of the hotspot region markers and tree-based species identification.

No.	Region Name	Pi	Alignment Length (bp)	ML Tree (%)	BLAST (%)	Total Identification Rate (%)
1	petG-trnW-CCA	0.091	175	75	83.3	79.2
2	rpl32-trnL-UAG	0.083	885	83.3	83.3	83.3
3	rpl16-rps3	0.073	217	83.3	83.3	83.3
4	rps8-rpl14	0.069	335	91.7	91.7	91.7
5	rps16-trnQ-UUG	0.067	1237	83.3	83.3	83.3
6	ndhG-ndhI	0.062	409	83.3	100	91.7
7	petN-psbM	0.059	1327	91.7	100	95.9
8	accD-psaI	0.055	819	83.3	83.3	83.3
9	trnG-GCC-trnfM-CAU	0.054	209	83.3	83.3	83.3
10	atpF-atpI	0.053	1956	91.7	100	95.9
11	trnT-UGU-trnL-UAA	0.052	995	83.3	83.3	83.3
12	psbZ-trnG-GCC	0.052	424	83.3	83.3	83.3
13	trnK-UUU-rps16	0.051	565	83.3	83.3	83.3
14	ndhC-trnV-UAC	0.049	1441	100	100	100.0
15	trnT-GGU-psbD	0.041	1589	91.7	100	95.9
16	psbE-petL	0.040	1274	100	100	100.0

**Table 3 genes-14-01149-t003:** Sequence and tree-based species identification rate of existing barcode markers in the genus *Ranunculus*.

No.	Region Name	Length (bp)	ML Tree (%)	BLAST (%)	Total Identification Rate (%)
1	*matK*	1527	83	100	91.5
2	*rbcL*	1428	75	75	75
3	*psbA-trnH-GUG*	366	75	75	75
4	*trnL-UAA-trnF-GAA*	418	75	75	75

**Table 4 genes-14-01149-t004:** Sequence and tree-based species identification rate of existing barcode markers in the genus *Ranunculus*.

No.	Intergenic Region	Forward Primer (5′–3′)	Reverse Primer (5′–3′)	Product Size Range (bp)	Annealing Temperature (°C)	Sequencing Success (%)
1	petG-trnW-CCA	TACAGACGCGGTGATCAGTTGGAC	CCAAAACCCGATGTCGTAGGTTC	200–250	58	100
2	rpl32-trnL-UAG	GCAAAATCTATTTCCACCGGGAAT	GTCTACCGATTTCACCATAGCG	800–900	58	100
3	rpl16-rps3	CAACGAGTCACACACTAAGCA	ACTATCTATGGGGCATTAGGAA	200–300	58	100
4	rps8-rpl14	TCCATGTCAGCATTTCGTATCG	GTGCAATCGCTCGAGAGTTGA	370–410	58	100
5	rps16-trnQ-UUG	CGCACGTTGCTTTCTACCACA	CGAATCCTTCCGTCCCAGAG	1000–1200	58	100
6	ndhG-ndhI	GGTCGGTTACCAATGTCAGTGA	AAGGAGCTGTGCAGCAGCGA	600–700	58	100
7	petN-psbM	GGCTGCTTTAATGGTAGTCTTTAC	TCGCATTCATTGCTACTGCACTG	1200–1300	58	100
8	accD-psaI	GAGTGAGTTATTTCAGCTTCACG	GGAGGGTAAGTTGAAAGTTGTCAT	750–850	58	100
9	trnG-GCC-trnfM-CAU	CGATTCCCGCTATCCGCCTA	GGTAGCTCGCAAGGCTCATAAC	250–300	58	100
10	atpF-atpI	GGCCAGTGACCCAAGGAAAC	TAGGGGAATCCATGGAGGGTCA	300–1900	58	100
11	trnT-UGU-trnL-UAA	CGGCTATCGGAATCGAACCG	GCGTCTACCAATTTCGCCATATC	800–1000	58	100
12	psbZ-trnG-GCC	CCCGTTGTATTTGCTTCTTCTGA	CCGCGTCTTCTCCTTGGCAA	500–600	58	100
13	trnK-UUU-rps16	AGCCGCACTTAAAAGCCGAGTA	CGATCCCGAAGAGAGGGAAG	500–700	58	100
14	ndhC-trnV-UAC	GAGTTTCTCTGGCCCTTCATTA	TACCGAGAAGGTCTACGGTTC	1300–1400	58	100
15	trnT-GGU-psbD	GGCGTAAGTCATCGGTTCAAAT	TCTCCGTAACCAGTCATCCATA	1500–1600	58	100
16	psbE-petL	CGTGCTTCCAGACATGCTGA	GCCGTATCTTGCTCAGACCAAT	1200–1300	58	100

## Data Availability

The genome sequence data that support the findings of this study are openly available from NCBI GenBank (https://www.ncbi.nlm.nih.gov/ accessed on 2 March 2022) under the accession no. ON755204.

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
