# Peer review of "Complete Chloroplast Genome Determination of Ranunculus sceleratus from Republic of Korea (Ranunculaceae) and Comparative Chloroplast Genomes of the Members of the Ranunculus Genus"

_genes, 2023, doi:10.3390/genes14061149_

Round 1

Reviewer 1 Report

The manuscript titled “Complete Chloroplast Genome Determination of Ranunculus sceleratus from Korea (Ranunculaceae) and Comparative Chloroplast Genomes of the Members of the Ranunculus Genus” aim to sequence the chloroplast genome of R. sceleratus from Korea, and chloroplast sequences were compared and analyzed among Ranunculus species. Comparison of the Ranunculus genomes provides useful information regarding species identification and evolution that will guide future phylogenetic analyses. I considered the manuscript should also make following improvements.

 1.In “Materials and Methods- 2.1. Plant Sampling and Sequencing”, who collected and identified the species, recommended to be added.

2.In “Materials and Methods- 2.2. Chloroplast Genome Assembly and Annotation”, Reference gene for species annotation, recommended to be added.

3.     “To construct phylogenetic trees, 13 cp genomes of the species belonging to the family Ranunculaceae were downloaded from NCBI GenBank.” How did authors confirm these cp sequences are right?

4.     “Moreover, 100 bootstrap replicates were included for the ML tree.” I think 1000 bootstrap replicates seem more accurate?

5.     “The GC content of the cp genome was 37.9%, which corresponded with that of R. sceleratus from China; twelve other Ranunculus species genomes were reported to have a similar GC content (37.7–37.9%; Supplementary Table S2).” It is recommended to compare the GC content of each region. Such as GC content in LSC region(%), GC content in IR region(%), GC content in SSC region(%).

6.     The manuscript including 1. Introduction, 2. Materials and Methods, 3. Results and Discussion, there seems to be no clear conclusion, recommended to add and include future recommendations in the conclusion.

7.     “Figure 6. Nucleotide diversity in the entire chloroplast genome of 11 species. (a) coding sequence;(b) intergenic region.” A clearer picture is recommended.

The article should make minor revisions of English / grammatical.

Reviewer 2 Report

Kim et al. sequenced the complete chloroplast genome of Ranunculus sceleratus and identified the potential barcode markers for Ranunculus. I have some comments as following.

1. There are so many figure and table in the MS. I suggest that the authors move some of the figures and tables to additional information. For example, Table 2.

2. For Figure 5, Please give the tree with branch length.

3. “The combined 37 datasets showed a high species identification rate of 100% com-pared to the existing single marker”. I don't quite understand is that the authors are planning to concatenate these 37 genes as barcodes? If so many barcodes are used, then this doesn't make sense and the authors need to evaluate these barcodes to give the optimal combination.

Reviewer 3 Report

1). Manuscript ID: Genes-2393207

2). Manuscript Title: Complete Chloroplast Genome Determination of Ranunculus sceleratus from Korea (Ranunculaceae) and Comparative Chloroplast Genomes of the Members of the Ranunculus Genus

3). General Comments

--Please follow the Journal format while revising the manuscript.

--Add scientific authority at the end of binomial names of all species when they are mentioned for the first time in the manuscript.

--Include full forms of all abbreviations/acronyms mentioned in the manuscript.
